# Efficacy and safety of tegoprazan in the treatment of gastroesophageal reflux disease: A protocol for meta-analysis and systematic review

Hanxue Zheng[1]☯, Shunqi Yuan[2]☯, Jianmin Liu[3]*

1 Department of Otorhinolaryngology Head and Neck Surgery, Deyang People's Hospital, Deyang Hospital Affiliated to Chengdu University of Traditional Chinese Medicine, Deyang, Sichuan, China, 2 Department of Otolaryngology Head and Neck Surgery, Longquanyi Hospital of West China Hospital of Sichuan University (The First People's Hospital of Longquanyi District of Chengdu), Chengdu, China, 3 Thyroid -ENT Head and Neck Surgery Department, Radiation Oncology Key Laboratory of Sichuan Province, Sichuan Clinical Research Center for Cancer, Sichuan Cancer Hospital & Institute, Sichuan Cancer Center, Affiliated Cancer Hospital of University of Electronic Science and Technology of China, Chengdu, China

☯ These authors contributed equally to this work.

* liujianminly@163.com

**Data Availability Statement:** All relevant data are within the manuscript and its Supporting Information files.

## Abstract

### Objective

The incidence of gastroesophageal reflux disease (GERD) is increasing year by year, the clinical manifestations are complex and diverse, and the adverse effects of long-term use of proton pump inhibitors and gastrointestinal motility drugs have been of great concern in recent years. The effectiveness of tegoprazan in the treatment of GERD is still controversial. This protocol describes a systematic review and meta-analysis to evaluate the efficacy and safety of tegoprazan in the treatment of gastroesophageal reflux disease.

### Methods

PubMed, Embase, Cochrane Library and Web of Science will be searched from the database inception to 1 March 2023. All randomized controlled trials related to tegoprazan for GERD will be included. Extracted data will include publication details, basic information, demographic data, intervention details and patient outcomes. The primary outcome will be complete resolution of major symptoms, complete resolution of heartburn, proportion of heartburn-free days, chronic cough, hoarseness, and adverse events. Risk of bias will be assessed using the Cochrane Collaboration's tool for assessing risk of bias. Article selection, data extraction and risk of bias assessment will be performed in duplicate by two independent reviewers. If the meta-analysis is precluded, we will conduct a descriptive synthesis using a best-evidence synthesis approach.

**Funding:** The author(s) received no specific funding for this work.

**Competing interests:** The authors have declared that no competing interests exist.

**Abbreviations:** GERD, Gastroesophageal reflux disease; LES, lower esophageal sphincter; SMD, Standard mean difference; RCTs, randomized control trials; WMD, weighted mean difference; CI, Confidence Interval; RR, the risk ratio; OR, odds ratio.

## Discussion

The results of this study will provide reliable evidence to evaluate the efficacy and safety of tegoprazan in the treatment of GERD and help patients, physicians and clinical investigators choose the most appropriate treatment.

## Background

Gastroesophageal reflux disease (GERD) is a disease in which the contents of the stomach or duodenum flow back into the esophagus causing discomfort and/or complications, both physiologic and pathologic [1]. Typical symptoms are esophageal manifestations, with heartburn and reflux being the most characteristic, as well as chest pain and dysphagia [2, 3]. Extra-esophageal manifestations, such as laryngitis, chronic cough and asthma, and even hysteria such as foreign body sensation and blockage in the pharynx, can also be present [4]. In Asia, heartburn or reflux is diagnosed at least once a week, and then the endoscopic presentation of the esophagus is divided into three types: reflux esophagitis, non-erosive reflux disease and Barrett's esophagitis [5]. The prevalence of GERD is increasing year by year, as shown by the recent publications [6, 7]. The pathogenesis of GERD is complex, and current treatment is based on the use of proton pump inhibitors to promote esophageal mucosal healing and gastrointestinal motility drugs to increase lower esophageal sphincter (LES) pressure [8]. Since the recurrence rate of the disease is 57%-90%, patients must take proton pump inhibitors and prokinetic drugs to relieve their symptoms for a long time [9]. Drugs such as histamine H2 receptor blockers and proton pump inhibitors (PPIs), which inhibit gastric acid secretion, are effective in the treatment of acid-related diseases and may improve patients' quality of life [10]. However, there are still some limitations of the existing drug therapy. Some adverse effects of long-term proton pump inhibitors and prokinetic drugs have received attention in recent years: long-term use of proton pump inhibitors has led to the development of gastric adenocarcinoma, osteoporosis in the elderly, and bacterial overgrowth in the intestinal tract [11, 12], and the 5-hydroxytryptamine-4 agonist tegaserod has been discontinued because of the increase in serious cardiovascular system adverse effects. On the other hand, duodenal gastroesophageal reflux disease is a complex reflux disease with independent risk factors, and proton pump inhibitors are not effective in its treatment [13]. In recent years, a new potassium-competitiveacid blocker (P-CAB) has gained much attention because it can better overcome the disadvantages of PPIs. Unlike PPIs, P-CAB is a new type of highly efficient and selective gastric H+/K+-ATPase inhibitor that does not require activation in a strong acid environment to function. The greatest advantage of tegoprazan, a novel P-CAB, is its ability to bind to the K + binding site of H+/K+-ATPase in a competitive and reversible manner without acid activation and without any conversion, exhibiting faster gastric acid inhibition than PPIs and almost complete inhibition of gastric acid secretion [9, 14]. In addition, P-CAB-induced gastric acid inhibition is not affected by the ab initio synthesis of proton pumps in gastric lining cells, thus enabling faster maximal inhibition and longer duration of action, making P-CAB a novel, highly selective and efficient acid inhibitor [15, 16]. We present the first systematic review and meta-analysis protocol based on randomized controlled trials to investigate the efficacy and safety of tegoprazan in the treatment of gastroesophageal reflux disease.

## Material and methods

### Protocol and registration

This protocol needs to be reported following the guidelines of the Cochrane Handbook for Systematic Reviews of Interventions and the Preferred Reporting Items for Systematic Reviews

and Meta-Analyses. The study is expected to begin on 1 March 2023 and end on 1 May 2023. The review will be conducted in accordance with the PRISMA guidelines (S1 Checklist) [17]. This protocol was registered in PROSPERO and registration number is CRD42023408904 (S1 File).

## Eligibility criteria

Inclusion criteria: for adults diagnosed with gastroesophageal reflux disease [18], the experimental group was treated with tegoprazan, and the control group was treated with placebo or blank intervention, the primary outcome was complete resolution of major symptoms, complete resolution of heartburn, proportion of heartburn-free days, adverse events, this study will only consider randomized control trials (RCTs).

Exclusion criteria: the following studies will be excluded: duplicate articles, meta-analyses, reviews, protocols, animal experiments, letters, full text not available.

## Search strategy

The following electronic databases: PubMed, Embase, Cochrane Library and Web of Science will be searched from inception to April 20, 2023. In addition, we will review the list of references, relevant conference literature and trial registry database (WHO International Clinical Trials Registry Platform and Clinical Trials.gov) to identify additional studies.

The search strategy is shown in S2 File. The following search terms will be used singly or as combinations (Mesh terms and free words): tegoprazan, gastroesophageal reflux.

## Data collection and analysis

**Studies screening process.**   Two reviewers are required to independently screen the retrieved studies. Briefly, they will exclude duplicate studies and studies that do not meet the inclusion criteria by reading the titles and abstracts. They will read the full text of each study to select those that meet the inclusion criteria. Any disagreements will be resolved through discussion with a third reviewer. The entire study selection process is shown in Fig 1. To obtain missing data, we will contact the first or corresponding author of the trials via email. To determine whether the evaluation results are inconsistent, we will conduct intent analysis and sensitivity analysis on the data of all participants.

**Data extraction.**   Data extraction from the included studies will be done independently by two reviewers following a data acquisition list [19]. The list will include the basic information (author, title, journal, year of publication and country of publication), study design (study size, randomization, allocation concealment, blinding methods, dose of interventions and treatment duration), outcome measures and conflicts of interest. If necessary, a third reviewer will double-check the data to ensure consistency.

If data is missing or incomplete in any study, we will contact the authors to obtain these data. In the absence of available data, we deleted the study.

**Assessment of risk of bias.**   The risk of bias in the included studies will be determined by two reviewers independently using the Cochrane risk of bias tool (RoB2.0) [20]. Which has five domains (randomization process, deviations from intended interventions, missing outcome data, measurement of the outcome, and selection of the reported result) which is judged as "low risk", "some concern" or "high risk" of bias. The response options for an overall risk-of-bias judgment are the same as for individual domains. If any disagreements, the risk assignment will be settled through arbitration of a third reviewer.

**Measures of treatment effect.**   The Rvman5.4 software was used for statistical analysis, and the stata15.0 software was used for the evaluation of sensitivity and publication bias. For

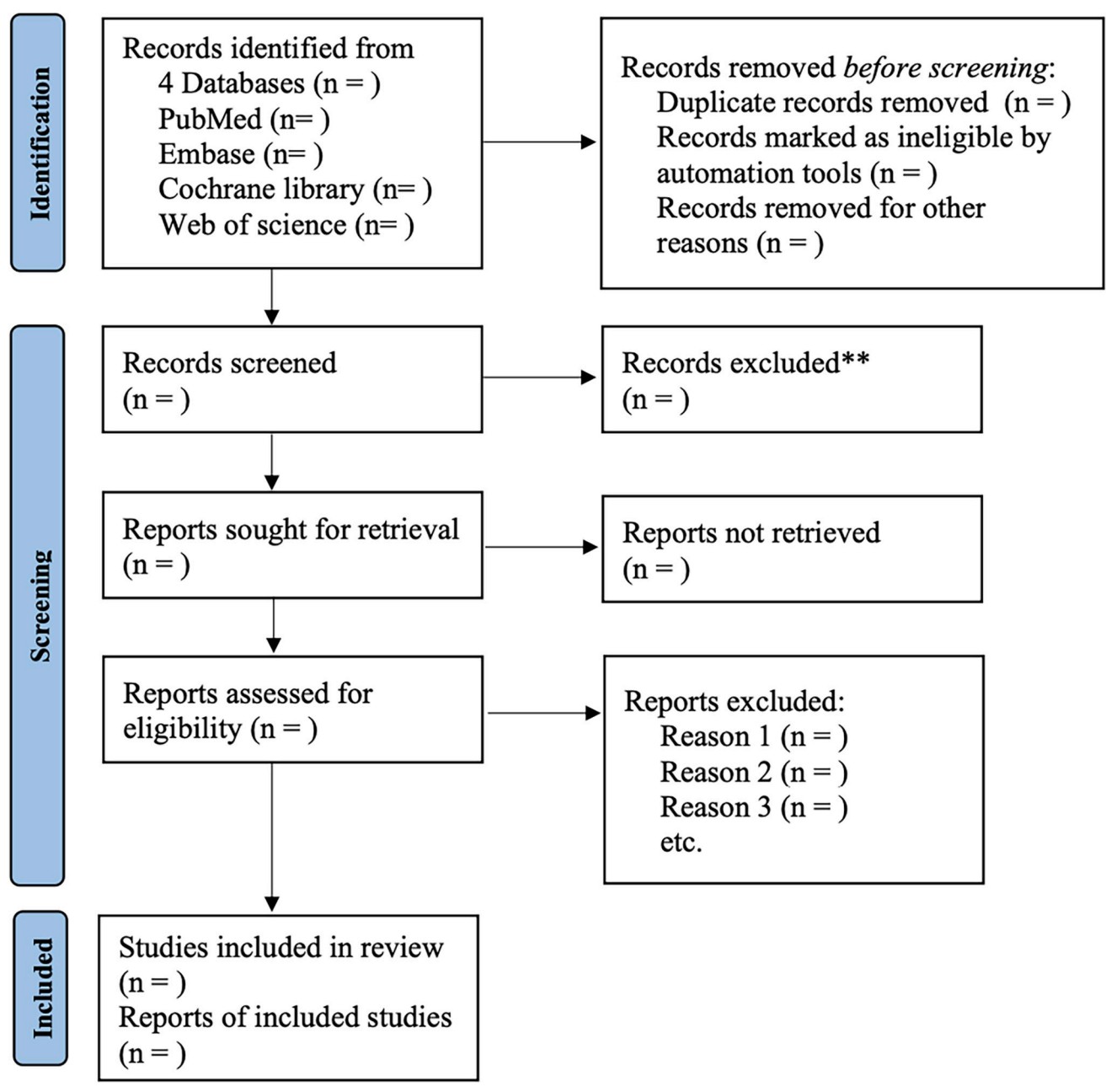

**Fig 1. Flow diagram of the study design.**

continuous outcome data, the Standard mean difference (SMD) or weighted mean difference (WMD) with 95% CI (Confidence Interval) will be used. For dichotomous data, the risk ratio (RR) or odds ratio (OR) with 95% CI will be used for analysis.

**Assessment of heterogeneity.** We will evaluate heterogeneity using the Cochrane's Q test and I2 statistics (p>0.05 for Q statistics and I2 <50% indicating statistical homogeneity) [21]. Because studies may include different dose of tegoprazan, meta-analyses will be performed using random-effects models. A subgroup analysis will be performed to explore the possible causes. If the heterogeneity is greater than 75%, a meta-analysis will not be performed. We will provide a narrative, qualitative summary.

**Assess the quality of the evidence.** The internationally accepted grading of recommendations assessment development and evaluation (GRADE) [22] was used to grade the quality of evidence for outcomes. All RCTs were included in this study. RCTs were set as the highest level of evidence, and there were 5 factors that could reduce the quality of evidence. The quality of the evidence: study limitations, inconsistent findings, indirectness of findings, imprecision of findings, publication bias.

**Assessment of publication biases.** Publication bias was tested for inclusion of more than 10 studies using funnel plots and Egger's test [23]. If the two sides of the funnel plot are asymmetrical, it means that there is a high possibility of publication bias, and Egger's test can be used. If the P value is < 0.05, it means that there is a publication bias, and if p> 0.05, it is the opposite of the funnel plot result, at this time we can further use the cut and complement method to further determine.

## Subgroup analysis

If data are available, subgroup analysis will be performed to assess the heterogeneity according to the dose of tegoprazan, treatment time, age, gender.

## Sensitivity analysis

If possible, sensitivity analysis will be used to evaluate how uncertain assumptions of data and usage affect the robustness of the combined results. We will judge the specific impact of an article on the results of statistical analysis by eliminating literature one by one [24].

## Ethics and dissemination

This review will not require ethical approval as it does not infringe on anyone's interests. The results will be published in a peer-reviewed journal or disseminated through conferences.

## Discussion

This systematic review will build on previous randomized controlled studies of tegoprazan for the treatment of gastroesophageal reflux disease, and the conclusions drawn from this systematic review will be beneficial to patients and clinicians with gastroesophageal reflux disease.

The pathophysiological basis of GERD is the damage to the esophageal mucosa caused by hydrogen ions in the reflux and pepsin activated in an acidic environment, which further penetrates the esophageal mucosa and stimulates nociceptive receptors, leading to the typical GERD disorder [25]. PPI acid-suppression therapy can control the pH of the reflux and thus has a significant efficacy in GERD [26, 27]. Nevertheless, about 30%-40% of GERD patients do not have good control of heartburn and acid reflux after standard dose PPI therapy, and about 20%-30% of RE patients fail to achieve healing of esophagitis [28, 29]. A randomized, double-blind, placebo-controlled, multicenter phase III clinical trial conducted in Korea compared the efficacy, safety, and tolerability of tegoprazan with placebo for the treatment of NERD [30]. The primary endpoint was the rate of complete remission of major symptoms (heartburn and regurgitation) at week 4 of treatment. The 324 patients with NERD were randomly divided into the tegoprazan 50 mg group, the tegoprazan 100 mg group, and the placebo group, all administered once daily for 4 weeks. The results showed that the rates of complete remission of major symptoms at week 4 were 42.5% (45/106), 48.5% (48/99) and 24.2% (24/99) in the tegoprazan 50 mg, 100 mg and placebo groups, respectively, with the tegoprazan group outperforming the placebo group (P = 0.005 8 and P = 0.000 4, respectively). The mean percentages of complete heartburn relief and time without heartburn were also higher in the tegoprazan

group than in the placebo group (both P<0.05). This study showed that tegoprazan 50 and 100 mg showed better results compared to placebo for NERD. All three clinical trials [30–32] reported treatment emergent adverse events (TEAEs), most of which were mild and tolerated by patients. The more common TEAEs for tegoprazan were mainly gastrointestinal adverse events (2% ~ 4.9%) and headache (1% ~ 4.9%), with no significant differences compared with positive controls or placebo. In addition, one case of elevated alanine aminotransferase in the tegoprazan 50 mg and 100 mg groups, and one case of elevated aspartate aminotransferase in the tegoprazan 100 mg and lansoprazole groups, respectively, were reported in efficacy trials investigating treatment, suggesting the need to pay attention to patients' liver function during treatment [22]. However, the sample size of the clinical trials conducted so far is limited, and the safety remains to be observed in more subsequent clinical trials.

The protocol has several advantages. We plan to search multiple English databases to ensure a comprehensive search of the literature. Any meta-analysis will be conducted in accordance with the Cochrane Handbook for Systematic Reviews of Interventions. Another advantage is that strict eligibility criteria will be used to ensure the quality of the included randomized controlled trials. In addition, complete resolution of major symptoms was selected as a target outcome because it is an important indicator of treatment of GERD.

## Supporting information

**S1 Checklist. Prisma checklist.**
(PDF)

**S1 File. PROSPERO registration document.**
(PDF)

**S2 File. Retrieval strategy.**
(DOCX)

## Author Contributions

**Conceptualization:** Hanxue Zheng.

**Data curation:** Hanxue Zheng.

**Formal analysis:** Hanxue Zheng, Shunqi Yuan, Jianmin Liu.

**Funding acquisition:** Hanxue Zheng.

**Investigation:** Hanxue Zheng, Jianmin Liu.

**Methodology:** Hanxue Zheng.

**Project administration:** Hanxue Zheng, Shunqi Yuan.

**Resources:** Hanxue Zheng, Jianmin Liu.

**Software:** Hanxue Zheng.

**Supervision:** Hanxue Zheng, Shunqi Yuan, Jianmin Liu.

**Validation:** Hanxue Zheng.

**Visualization:** Hanxue Zheng, Shunqi Yuan, Jianmin Liu.

**Writing – original draft:** Hanxue Zheng, Jianmin Liu.

**Writing – review & editing:** Hanxue Zheng, Jianmin Liu.

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
