## [Decision Letter · Decision Letter 0]

20 Feb 2024

PONE-D-23-42637Efficacy and safety of tegoprazan in the treatment of gastroesophageal reflux disease: A protocol for meta-analysis and systematic reviewPLOS ONE

Dear Dr. Liu,

Thank you for submitting your manuscript to PLOS ONE. After careful consideration, we feel that it has merit but does not fully meet PLOS ONE’s publication criteria as it currently stands. Therefore, we invite you to submit a revised version of the manuscript that addresses the points raised during the review process.

We look forward to receiving your revised manuscript.

Kind regards,

Dong Keon Yon, MD, FACAAI, FAAAAI

Academic Editor

PLOS ONE

Journal Requirements:

5. We note that this manuscript is a systematic review or meta-analysis; our author guidelines therefore require that you use PRISMA guidance to help improve reporting quality of this type of study. Please upload copies of the completed PRISMA checklist as Supporting Information with a file name “PRISMA checklist”.

Additional Editor Comments:

Thank you for submitting your manuscript. The reviewers and I believe it is of potential value for our readers. However, the reviewers have raised a number of very important issues, and their excellent comments will need to be adequately addressed in a revision before the acceptability of your manuscript for publication in the Journal can be determined. We cannot guarantee that your revised paper will be chosen for publication; this would be solely based on how satisfactorily you have addressed the reviewer comments.

# Ref 16 is too old guideline. Please change and cite the reference below.

DOI: https://doi.org/10.54724/lc.2022.e9

Reviewers' comments:

Reviewer's Responses to Questions

**Comments to the Author**

1. Does the manuscript provide a valid rationale for the proposed study, with clearly identified and justified research questions?

Reviewer #1: No

Reviewer #2: Yes

2. Is the protocol technically sound and planned in a manner that will lead to a meaningful outcome and allow testing the stated hypotheses?

Reviewer #1: Partly

Reviewer #2: Yes

3. Is the methodology feasible and described in sufficient detail to allow the work to be replicable?

Reviewer #1: No

Reviewer #2: Yes

4. Have the authors described where all data underlying the findings will be made available when the study is complete?

Reviewer #1: No

Reviewer #2: Yes

5. Is the manuscript presented in an intelligible fashion and written in standard English?

Reviewer #1: No

Reviewer #2: No

6. Review Comments to the Author

You may also provide optional suggestions and comments to authors that they might find helpful in planning their study.

Reviewer #1: This protocol needs to be reported following the guidelines of the Cochrane Handbook for Systematic Reviews of Interventions and the Preferred Reporting Items for Systematic Reviews and Meta-Analyses; the authors need to elucidate the necessity of performing such meta-analysis, as some similar reviews have been reported; and the manuscript needs extensive revision for language and grammar.

Reviewer #2: This is a good article, but there are some part that needs to be modified.

1. please put your country in your affiliation statement

2. please make same bold letters in subtitle of Abstract. (e.g. Objective: Methods: ,etc,)

3. please put PubMed, Embase, Cochrane Library databases, and Web of Science

- you have to put , before using "and"

4. Authors have to put line numbers in manuscript for readability of reviewers

5. Could you put reference in following sentences: "Drugs such as histamine H2 receptor blockers and proton pump inhibitors (PPIs), which inhibit gastric acid secretion, are effective in the treatment of acid-related diseases and may improve patients' quality of life. However, there are still some limitations of the existing drug therapy."

6. MATERIAL AND METHODS/ Patients

You don't have to mention this part. if you need to mention this part. Please explain it.

7. Do you have a reference for standard of Inclusion criteria and exclusion criteria that you utilized in your article.

8. I know what you want to talking about in "Studies collection" this section, but could you please explain it in an easy-to-understand manner?

9. Could you put more references in Methods part.

It would be helpful, you could get many references about your methods section.

# Sandbank M, Bottema-Beutel K, Crowley LaPoint S, Feldman JI, Barrett DJ, Caldwell N, Dunham K, Crank J, Albarran S, Woynaroski T. Autism intervention meta-analysis of early childhood studies (Project AIM): updated systematic review and secondary analysis. BMJ. 2023 Nov 14;383:e076733. doi: 10.1136/bmj-2023-076733. PMID: 37963634; PMCID: PMC10644209.

10. Discussion part should divided in results part and discussion part

please refer this article

Hume-Nixon M, Quach A, Reyburn R, Nguyen C, Steer A, Russell F. A Systematic Review and meta-analysis of the effect of administration of azithromycin during pregnancy on perinatal and neonatal outcomes. EClinicalMedicine. 2021 Sep 9;40:101123. doi: 10.1016/j.eclinm.2021.101123. PMID: 34541478; PMCID: PMC8436060.

11. Founding to Funding

12. please modify reference with "Times new roman, 12pt"

13. make whole manuscript of line spacing with 160%

7. PLOS authors have the option to publish the peer review history of their article (what does this mean?). If published, this will include your full peer review and any attached files.

Reviewer #1: No

Reviewer #2: No

---

## [Author Response · Author response to Decision Letter 0]

16 Mar 2024

Dear editor and dear reviewers

Re Manuscript ID (PONE-D-23-42637) entitled " Efficacy and safety of tegoprazan in the treatment of gastroesophageal reflux disease: A protocol for meta-analysis and systematic review ". Thank you for your letter and for the reviewer’s comments. Those comments are all valuable and very helpful for revising and improving our paper, as well as the important guiding significant to our research. We have studies comments carefully and have made correction which we hope meet with approval. Revised portion are marked in red in the paper. The main correction in the paper and responds to the reviewer’s are as flowing:

Additional Editor Comments:

Thank you for submitting your manuscript. The reviewers and I believe it is of potential value for our readers. However, the reviewers have raised a number of very important issues, and their excellent comments will need to be adequately addressed in a revision before the acceptability of your manuscript for publication in the Journal can be determined. We cannot guarantee that your revised paper will be chosen for publication; this would be solely based on how satisfactorily you have addressed the reviewer comments.

# Ref 16 is too old guideline. Please change and cite the reference below.

DOI: https://doi.org/10.54724/lc.2022.e9

Responds: Many thanks to the editors for their comments, and we have revised the article's 16th reference.

Reviewer #1: This protocol needs to be reported following the guidelines of the Cochrane Handbook for Systematic Reviews of Interventions and the Preferred Reporting Items for Systematic Reviews and Meta-Analyses; the authors need to elucidate the necessity of performing such meta-analysis, as some similar reviews have been reported; and the manuscript needs extensive revision for language and grammar.

Responds: Thank you very much for the reviewer's comments, We followed the Cochrane Handbook for Systematic Reviews of Interventions and the Preferred Reporting Items for Systematic for articles Reviews and Meta-Analyses were revised to add study necessity. However, our search found no similar review assessment.

Reviewer #2: This is a good article, but there are some part that needs to be modified.

1. please put your country in your affiliation statement

responds: Many thanks to the reviewers, we have added the countries.

2. please make same bold letters in subtitle of Abstract. (e.g. Objective: Methods: ,etc,)

responds: Many thanks to the reviewers, we have revised it.

3. please put PubMed, Embase, Cochrane Library databases, and Web of Science

- you have to put , before using "and"

responds: Many thanks to the reviewers, we have revised it.

3. Authors have to put line numbers in manuscript for readability of reviewers

Responds: Thank you very much to the reviewer, we have added line numbers.

4. Could you put reference in following sentences: "Drugs such as histamine H2 receptor blockers and proton pump inhibitors (PPIs), which inhibit gastric acid secretion, are effective in the treatment of acid-related diseases and may improve patients' quality of life. However, there are still some limitations of the existing drug therapy."

Responds: Thank you very much to the reviewer, we have added reference.

6. MATERIAL AND METHODS/ Patients

You don't have to mention this part. if you need to mention this part. Please explain it.

responds: Many thanks to the reviewers for their comments, we have revised the article. This section has been deleted.

7. Do you have a reference for standard of Inclusion criteria and exclusion criteria that you utilized in your article.

responds: Our reference on inclusion criteria is at 18.

8. I know what you want to talking about in "Studies collection" this section, but could you please explain it in an easy-to-understand manner?

responds：The purpose of our paragraph is to express a process by which we sift through the articles.

9. Could you put more references in Methods part.

It would be helpful, you could get many references about your methods section.

responds: Many thanks to the reviewers for their comments, which we have revised.

# Sandbank M, Bottema-Beutel K, Crowley LaPoint S, Feldman JI, Barrett DJ, Caldwell N, Dunham K, Crank J, Albarran S, Woynaroski T. Autism intervention meta-analysis of early childhood studies (Project AIM): updated systematic review and secondary analysis. BMJ. 2023 Nov 14;383:e076733. doi: 10.1136/bmj-2023-076733. PMID: 37963634; PMCID: PMC10644209.

10. Discussion part should divided in results part and discussion part

please refer this article

Hume-Nixon M, Quach A, Reyburn R, Nguyen C, Steer A, Russell F. A Systematic Review and meta-analysis of the effect of administration of azithromycin during pregnancy on perinatal and neonatal outcomes. EClinicalMedicine. 2021 Sep 9;40:101123. doi: 10.1016/j.eclinm.2021.101123. PMID: 34541478; PMCID: PMC8436060.

responds; The reviewer's comments are very much appreciated, as our article in this instance was a planner and did not have a results section, so we made suitable changes to the discussion.

11. Founding to Funding

responds: Many thanks to the reviewers for their comments, which we have revised.

12. please modify reference with "Times new roman, 12pt"

responds: Many thanks to the reviewers for their comments, which we have revised.

13. make whole manuscript of line spacing with 160%

responds: Many thanks to the reviewers for their comments, which we have revised.

---

## [Decision Letter · Decision Letter 1]

4 Apr 2024

Efficacy and safety of tegoprazan in the treatment of gastroesophageal reflux disease: A protocol for meta-analysis and systematic review

PONE-D-23-42637R1

Dear Dr. Liu,

We’re pleased to inform you that your manuscript has been judged scientifically suitable for publication and will be formally accepted for publication once it meets all outstanding technical requirements.

Kind regards,

Dong Keon Yon, MD, FACAAI, FAAAAI

Academic Editor

PLOS ONE

Additional Editor Comments (optional):

This is an excellent paper.

Reviewers' comments:

Reviewer's Responses to Questions

**Comments to the Author**

1. Does the manuscript provide a valid rationale for the proposed study, with clearly identified and justified research questions?

Reviewer #2: Yes

2. Is the protocol technically sound and planned in a manner that will lead to a meaningful outcome and allow testing the stated hypotheses?

Reviewer #2: Yes

3. Is the methodology feasible and described in sufficient detail to allow the work to be replicable?

Reviewer #2: Yes

4. Have the authors described where all data underlying the findings will be made available when the study is complete?

Reviewer #2: Yes

5. Is the manuscript presented in an intelligible fashion and written in standard English?

Reviewer #2: Yes

6. Review Comments to the Author

You may also provide optional suggestions and comments to authors that they might find helpful in planning their study.

Reviewer #2: Congratulations. The manuscript is much better than the original version.

Actually, I think there have to be more references in the methods section.

please put more references in your methods section while proof process

7. PLOS authors have the option to publish the peer review history of their article (what does this mean?). If published, this will include your full peer review and any attached files.

Reviewer #2: No
